# Lung Function Changes with Swim Training in Healthy and Allergic Endurance Athletes

**DOI:** 10.3390/jfmk10020231

**Published:** 2025-06-18

**Authors:** João Rodrigues, Bárbara Jesus, Paulo Caseiro, António Jorge Ferreira, Luís Rama

**Affiliations:** 1Faculty of Sports Sciences and Physical Education, CIPER, University of Coimbra, Avenida de Conímbriga, Santa Clara, Pavilhão 3, 3040-248 Coimbra, Portugal; jmrodrigues@live.com.pt; 2Faculdade de Motricidade Humana, CIPER, University of Lisbon, Estrada da Costa, Cruz Quebrada-Dafundo, 1499-002 Lisbon, Portugal; 3Faculty of Economics, CEISUC, University of Coimbra, Avenida Dias da Silva 165, 3004-512 Coimbra, Portugal; 4Health Technology School, Polytechnic Institute of Coimbra, Rua 5 de Outubro, 3045-043 Coimbra, Portugal; b4jesusjsb@gmail.com (B.J.); pcaseiro@estescoimbra.pt (P.C.); 5Faculty of Medicine, University of Coimbra, Azinhaga de Santa Comba, Pólo III, 3000-548 Coimbra, Portugal; aferreira@fmed.uc.pt

**Keywords:** swimming, spirometry, position, environment, allergy, asthma

## Abstract

Background: It is thought that swimming might elicit remarkable chronic lung function improvements that were not observed in land-based sports. However, there is no consensus on whether this is mainly attributable to genetic predisposition or specific training. This study aimed to characterize athletes’ lung function according to their swimming experience. Methods: The sample consisted of 45 male athletes, including 15 swimmers, 15 triathletes, and 15 runners. Spirometry tests were conducted under four conditions: seated on land, prone on land, seated while immersed in water, and prone while immersed in water. The tests were performed on the pool deck and pool, with the order of conditions randomized for each participant. Conclusions: The results of this study do not support the idea that there is a window of opportunity for greater lung function adaptations during childhood due to swim training. The accumulated years of swim training are the reason for the increased measurements of FVC and FEV_1_ of swimmers. The immersed seated condition measures differences in lung function more accurately relative to swim training experience. Swim training appears to primarily enhance FVC in healthy athletes, while in allergic and asthmatic athletes, it mainly promotes improvements in FEV_1_/FVC and FEF_25–75%_.

## 1. Introduction

It is thought that swimming might elicit remarkable chronic lung function improvements not observed in land-based sports [1,2,3,4,5]. However, there is no consensus on whether this is mainly attributable to genetic predisposition or specific training [6,7,8,9,10,11]. Another unresolved issue is whether those respiratory adaptations to swimming-specific training have a window of opportunity, specifically during prepubescent age, that makes those adaptations impossible to acquire later in life [12,13,14,15,16,17,18,19]. These inconsistencies could be due to a methodological lack of ecology: no previous research has evaluated the influence of swimming experience on measurements of lung function performed in different positions and environments, which could better show specific adaptations to swim training due to the measurements being performed in the same exposure environment. Also, allergies and asthma, known for inducing changes in lung function, need to be accounted for as they can also be confounding factors in this topic.

Due to the specificity of swim training, swimmers tend to show greater adaptations in lung function for total lung capacity (TLC), functional reserve capacity (FRC), residual volume (RV), tidal volume (V_T_), vital capacity (VC), forced vital capacity (FVC), forced expiratory volume in one second (FEV_1_), and FEV_1_/FVC than land-based athletes [1,2,3,4,5]. Nonetheless, some research did not find differences between swimmers and land-based athletes in variables such as FVC, FEV_1_, FEV_1_/FVC, forced expiratory flow (FEF_25%_), FEF_50%_, FEF_75%_, peak expiratory flow (PEF), maximum voluntary ventilation (MVV), maximum inspiratory pressure (MIP), and maximum expiratory pressure (MEP) [2,9,19,20,21]. The discrepancies in results may derive from differences in exposure to training duration and intensity, years of experience, and genetic endowment among samples [5].

Another factor that could blunt conclusive results is the possibility of windows of opportunity for enhancing lung growth and function in response to training stimuli. Research supporting this theory postulates that training-induced lung growth is age-dependent, potentially greater or exclusive during younger ages [13,14,15,16]. For example, Zinman and Gaultier suggested that increases in lung volumes from swimming training were greater in females under the age of 12 years [17]. On the contrary, Bovard stated that his results supported the idea that a swim training season during a peak period of lung growth did not influence changes in lung growth or function [18]. Still, a significant amount of the previously referenced research was conducted with post-pubescent athletes, indicating that respiratory adaptations to swim training can occur and suggesting that increased lung growth and function may also occur after somatic growth, provided specific conditions are met.

One last possible confounding variable is swimmers’ respiratory health. Swimmers are, by far, the athletes with the most prevalence of respiratory symptoms and pathologies [22,23]. This is due to a combination of factors: high ventilation (V_e_), changes in air temperature and humidity, and, most importantly, chronic exposure to high chlorine levels [23,24,25,26]. Chlorine interacts with organic matter to form chemicals that irritate the airway epithelium, producing inflammation and, in the long term, airway epithelium damage and remodeling [3,22]. Allergy symptoms derived from chlorine exposure and cold air may not usually have detectable sensitization in skin prick tests, i.e., not being IgE-mediated [27].

The aim of this study was to characterize athletes’ lung function according to their swimming experience. As there is no consensus in the literature, this study adopted a different methodological approach compared to previous research, aiming to clarify whether swimming-specific training improves lung function, whether any improvements are specific to the prone position and/or immersion, and whether the age at which swim training begins influences these improvements. The objectives of this study were to (1) compare lung function in different body positions (seated and prone positions) and environments (on land and in immersion) of adult athletes with different swimming-training start ages; (2) assess the impact of swimming experience on respiratory variables under two different conditions: the starting age of swim training and the number of years of swim training experience; and (3) evaluate how allergies and asthma affect lung function in athletes with varying levels of swimming experience. It was hypothesized that years of swimming experience could be associated with increases in FVC and FEV_1_, particularly in the prone immersed condition. Additionally, it was expected that athletes who began swimming during childhood would exhibit greater FVC and FEV_1_ values than those who started swimming as adults. We also hypothesized that asthmatic athletes, but not allergic or healthy ones, would show lower values of FEV_1_ and FEF_25–75%_ outside the water; however, these values would improve with the aid of hydrostatic pressure.

## 2. Materials and Methods

### 2.1. Design

This study had a cross-sectional design. Data was collected from swimming, triathlon, and running athletes. Swimming, triathlon, and running athletes willing to perform the spirometry measurements were gathered from clubs closest to the data collection location. The same technician used the same spirometer to conduct all spirometry tests in the same 25 m pool and pool deck.

First, the athletes completed an online questionnaire to gather age, sports background, and allergy and asthma diagnosis data. Second, about one week before data collection, all the athletes performed preliminary lung function tests to eliminate a possible learning effect error. For these tests, the order of the conditions in which each athlete performed them was standardized from what the technician reckoned to be the easiest to the most complex condition: seated position on land (SL), prone position on land (PL), immersed seated position (SI), and immersed prone position (PI). The protocol was set to help athletes better understand the spirometry tests before performing them in a prone position and during immersion.

Lastly, data collection was performed. Athletes’ anthropometric data were collected before any spirometry test. The order of the four different conditions in which each athlete performed the spirometry tests was randomized to eliminate possible respiratory muscle fatigue statistical error due to the maximum-effort nature of the tests. The best values of each variable were accepted for data analysis, or more trials were followed until an accepted trial was reached. The variables used in these tests were FVC, FEV_1_, FEV_1_/FVC, PEF, and FEF_25–75%_.

### 2.2. Sample

Given the study’s methodology, it was decided to focus solely on male athletes. This decision was driven by the limited number of female participants and the differences in allergy responses observed between sexes [28,29,30].

The sample was designed to comprise healthy male athletes involved in swimming, triathlon, or running who were aged between 20 and 55 years. Apart from allergies and asthma, athletes could not report having experienced any injury or illness episodes that impaired their regular training schedule within the three months preceding data collection, or any other medical contraindications. As athletes would later be grouped based on the age at which they started swimming, they were required to have at least 3 years of formal sports training before the age of 16. This ensured that any differences between groups would be attributed solely to the distinction between swimming and land-based sports, rather than a lack of sports experience during childhood [31,32]. Athletes were also required to have been actively involved in their current sport for at least 1 year when data collection began. Additionally, none of the athletes could be smokers. Online questionnaires were distributed to gather the athletes meeting the inclusion criteria.

The initial sample consisted of 48 male athletes. However, three were excluded from the final analysis: one athlete who, despite agreeing to participate, could not complete data collection due to scheduling conflicts; another who felt ill and was only able to complete the preliminary spirometry tests; and a third who exhibited significant spirometry deficits, according to the normal range, during the initial tests and was advised to undergo complementary medical exams. The final sample then comprised 45 male athletes: 15 swimmers (39.13 ± 11.03 years of age, 177.00 ± 6.72 cm of height, 75.37 ± 11.36 kg of body mass, and 22.93 ± 8.29 years of swim experience), 15 triathletes (42.33 ± 8.30 years of age, 176.53 ± 3.74 cm of height, 74.92 ± 8.38 kg of body mass, and 13.07 ± 8.40 years of swim experience), and 15 runners (40.87 ± 9.55 years of age, 176.47 ± 4.87 cm of height, 70.67 ± 4.09 kg of body mass, and 5.73 ± 8.29 years of swim experience). Of the athletes, 16 reported to have been diagnosed by a physician with some form of allergy and 9 reported being diagnosed with asthma.

### 2.3. Procedures

#### 2.3.1. Ethics

Questionnaire dissemination, reliability tests, and main tests were previously approved by the Ethics Committee from the Faculty of Sports Sciences and Physical Education of the University of Coimbra (CE/FCDEF-UC/00052022, CE/FCDEF-UC/000112023, and CE/FCDEF-UC/00092023, respectively). Consent to use the various facilities was obtained before any data collection. All participants were informed of the study protocol, assured of the confidentiality of their data, and provided with the opportunity to consent to participate voluntarily. They provided their consent digitally for the questionnaires and in writing for the respiratory data collection. The data used to generate the results of this article are available in a Mendeley Data database: DOI: 10.17632/rn2h8y8354.1.

#### 2.3.2. Questionnaires

Online questionnaires were disseminated using the LimeSurvey platform. To ensure data protection, identification codes were randomly generated and used instead of names to identify each athlete in questionnaires and studies. The questionnaires asked about athletes past sports experience; age at swim training onset; years of cumulative experience in formal swimming training (ST_years_), both before 16 years of age and overall; and weekly training session frequency, both relative to total training sessions (TSFreq_total_) and swimming-specific training sessions (TSFreq_swim_). Additionally, participants were asked to report any allergies and asthma that had been medically diagnosed and which exactly.

#### 2.3.3. Spirometry Data Collection

Before any spirometry data collection, anthropometric data were collected to normalize spirometric data. Height and body mass were collected. Before any tests, the questionnaires assessed sports experience and respiratory health. Air humidity, air temperature, and water temperature data were collected during each data collection period using a ThermoPro TP55 hygrometer (iTronics, Kowloon, Hong Kong). Before each data collection period, spirometer calibration was performed with the air humidity and temperature levels used for data collection using a certified 3 L syringe. Athletes were instructed not to take any asthma/allergy medications 12 h before data collection to prevent their effects from interfering with lung function variables.

Data from the spirometry tests were collected and analyzed by a specialized technician using a spirometer Spiropalm 6MWT (Cosmed, Rome, Italy) in a 25 m pool and pool deck. The spirometry test protocol adhered to the most current guidelines of the American Thoracic Society and the European Respiratory Society [33]. Before starting the tests, each participant was instructed on the spirometry test protocol. For the seated conditions (SL and SI), athletes kept a straight posture and placed their feet on the pool’s/pool deck’s floor at hip width, with hands resting on their thighs. For the SI condition, athletes used a height-adjustable bench, with measurements taken while immersed up to the collarbone level. At its shallowest point, the 25 m pool had a depth of 120 cm, where the athletes were seated during immersion. For the PI condition, athletes held onto the wall with their hands while the rest of the body was in the water, arms were extended at shoulder width, and the head was out of the water. Floating devices were positioned, as needed, under the abdominal region and the feet to help keep the prone position with only the shoulders, shoulder blades, and arms out of the water. The PL condition was the same as in the water: arms fully extended to the front at shoulder width. A soft mat (around 30 mm in thickness) was used to relieve the pressure in the athletes’ chests. The SL condition was the same as in the water, and athletes were sitting on the same bench set at the same height as in the water. During each data collection moment, every participant did at least 3 spirometric measurements, and after a careful analysis by the technician, the best values of each variable were accepted for data analysis, or more trials followed until an accepted trial was reached.

Spirometry data were collected through the Cosmed OMNIA 1.6.5 software (Cosmed, Rome, Italy). Spirometry data were analyzed using the raw data in liters (L) and liters per second (L/s), z-scores, and the percentage of the GLI (Global Lung Function Initiative)-predicted reference values [34] as provided by the software. All data regarding lung volumes are shown in BTPS following the most up-to-date recommendations of the American Thoracic Society (ATS) and European Respiratory Society (ERS) [33]. Preferably, raw lung function data were used for statistical analyses. However, in analyses of independent samples, z-scores and percentages of predicted reference values were used instead of raw data to normalize the data for anthropometric variables. During data collection, the average air humidity was 77.47%, the average air temperature was 29.05 °C, and the average water temperature was 30.38 °C.

#### 2.3.4. Sample Subdivisions

The diagnosis of asthma, as reported by athletes in the questionnaire (RAs), was used to divide athletes into two groups: those with a reported allergy (PRAs, *n* = 9) and those without a reported allergy (NRAs, *n* = 36). Additionally, the diagnosis of both allergic disease and asthma, also reported by athletes in the questionnaire (RA), was used to further divide athletes into two groups: those with a reported allergy (PRA, *n* = 16) and those without a reported allergy (NRA, *n* = 29). For the analysis of lung function values based on each athlete’s swim training onset, the sample was divided into two groups: Athletes with 3 or more years of formal swim training experience before 16 years of age were assigned to an early start group (ES, *n* = 27). The remaining athletes were assigned to the non-early start group (NES, *n* = 18). Years of cumulative experience in formal swimming training (ST_years_) were used to perform correlations with lung function variables.

#### 2.3.5. Statistical Analyses

Statistical analyses were performed using the IBM SPSS software, version 29.0 (IBM Corp, New York, NY, USA). The Shapiro–Wilk test was used to test the normality of data of every scale variable. Independent samples t-tests and one-way ANOVA tests were used to search for differences in variances between groups when a normal distribution was attained. Paired samples t-tests and repeated measures ANOVA tests were used to search for differences in variances between moments in the same samples when a normal distribution was attained. Mauchly’s test of sphericity was used for the repeated measures ANOVA tests.

Effect size magnitudes from these tests were calculated through Cohen’s D (d) and partial eta-squared (η^2^). For Cohen’s D, associations between variables were considered small when d < 0.5, medium when 0.5 ≤ d < 0.8, and large when d ≥ 0.8 [35]. Partial eta-squared (η^2^) was also used to estimate the effect sizes of the ANCOVA parametric tests. For the partial eta-squared, an association was considered small when η^2^ < 0.06, medium when 0.06 ≤ η^2^ < 0.14, and large when η^2^ ≥ 0.14 [36]. The Bonferroni post hoc test was used for the ANOVA tests to obtain in-between groups or moment differences.

Mann–Whitney tests for two variables and Kruskal–Wallis tests for more than two variables were used to search for differences in variances between groups when variables did not present a normal distribution. Wilcoxon signed-rank tests and Friedman’s tests were used to search for differences in variances between moments in the same samples when a normal distribution was attained. Pearson’s correlation (r) was used for samples with a normal distribution and Spearman’s correlation (r) for samples without a normal distribution. Correlations were considered weak when r < 0.3, moderate when 0.3 ≤ r < 0.5, and strong when r ≥ 0.5 [35].

## 3. Results

### 3.1. Lung Function Across Sports

When comparing across sports, no differences in TSFreq_Total_ were observed (Table 1), and no differences in lung function were observed under any studied conditions, regardless of whether using z-scores or the percentage of predicted reference values (Table 2). On the other hand, there were statistical differences in ST_years_ (*p* < 0.001) and TSFreq_swim_ (*p* < 0.001) (Table 1). Pairwise comparisons revealed that differences in ST_years_ were found exclusively between swimmers and runners (*p* < 0.001), while differences in TSFreq_swim_ were observed between swimmers and runners (*p* < 0.001) and between triathletes and runners (*p* < 0.001). The correlations of ST_years_ with FVC and FEV_1_ (Table 3) were significant and exclusive to the SI condition (r = 0.337, *p* = 0.023 and r = 0.340, *p* = 0.022, respectively, when analyzed using z-scores; r = 0.341, *p* = 0.022 and r = 0.347, *p* = 0.020, respectively, when analyzed using the percentage of predicted reference values).

### 3.2. Effects of Allergy and Asthma

When the sample was subdivided by the reported diagnosis of allergies and asthma (RA), no significant differences between groups were observed in either lung function variables or ST_years_. However, when ST_years_ were correlated with lung function variables, the NRA group showed a significant correlation only with FVC in the SI condition (r = 0.469, *p* = 0.010 using z-scores; r = 0.468, *p* = 0.010 using the percentage of predicted reference values). The PRA group showed positive correlations with FEV_1_/FVC_SL_ (r = 0.606; *p* = 0.013), FEF_25–75%SL_ (r = 0.576; *p* = 0.020), and FEF_25–75%PL_ (r = 0.583; *p* = 0.018) through the z-score values and positive correlations with FEV_1_/FVC_SL_ (r = 0.589; *p* = 0.016), FEV_1_/FVC_PL_ (r = 0.570; *p* = 0.021), FEF_25–75%SL_ (r = 0.594; *p* = 0.015), FEF_25–75%PL_ (r = 0.624; *p* = 0.010), FEF_25–75%SI_ (r = 0.571; *p* = 0.021), and FEF_25–75%PI_ (r = 0.555; *p* = 0.026) through the percentage of predicted reference values (Table 4).

When the sample was subdivided by a reported diagnosis of asthma (RAs), no significant differences were observed between groups in either lung function variables or ST_years_. However, when ST_years_ were correlated with lung function variables, the NRAs group showed a significant correlation only with FVC in the SI condition, both through z-score values (r = 0.380, *p* = 0.022) and the percentage of predicted reference values (r = 0.382, *p* = 0.022). The PRAs group showed positive correlations with FEV_1_/FVC in the SL condition (r = 0.850, *p* = 0.004), in the PL condition (r = 0.767, *p* = 0.016), and in the SI condition (r = 0.800, *p* = 0.010), as well as with FEF_25–75%_ in the SL condition (r = 0.717, *p* = 0.030), all based on z-score values. Additionally, using the percentage of predicted reference values, the PRAs group showed positive correlations with FEV_1_/FVC in the SL (r = 0.833, *p* = 0.005), PL (r = 0.710, *p* = 0.032), and SI (r = 0.758, *p* = 0.018) conditions, as well as with FEF_25–75%_ in the SL (r = 0.786, *p* = 0.012) and PL (r = 0.673, *p* = 0.047) conditions (Table 4).

### 3.3. Lung Function Differences Between Swim Training Onset

Subdivision of the sample into the early swim training onset (ES) and non-early swim training onset (NES) groups (Table 5) revealed a significant difference in FVC_PI_ (*p* = 0.035) between groups when analyzed using z-scores and significant differences in both FVC_SI_ (*p* = 0.045) and FVC_PI_ (*p* = 0.032) when analyzed using the percentage of predicted reference values. A significant difference in ST_years_ (*p* = 0.011) was also found. Due to this difference, ANCOVA analyses were conducted with ST_years_ as the covariate. These analyses revealed no significant differences between groups using z-scores or the percentage of predicted reference values.

## 4. Discussion

Given the previously reported heterogeneous results regarding improvements in lung function attributable to swimming-specific training [4,11,19,32,37,38,39], this study adopted a different approach aimed at clarifying this issue. Key factors, including body position and environment, age at the onset of swim training, and reported allergies and asthma diagnoses, were considered to better understand their individual effects on lung function.

Male endurance athletes from swimming, triathlon, and running were recruited for the sample, all of whom had previous experience with swimming training and had engaged in sports training for more than three years before the age of 16. These inclusion criteria were crucial, as exercise during prepubescent ages is believed to influence lung growth and function [13,14,15,16,17]. With these criteria, all athletes had exposure to sports during these ages, ensuring that any differences in lung function could be directly attributed to the age of onset and time spent in swim training, rather than a lack of sports training during the prepubescent years.

### 4.1. Homogenity of Lung Function Across Sports

Differences in lung function between sports were not found when analyzed by the z-score and the percentage of reference predicted values. Although not common, as in most of the literature points towards swimmers showing greater lung function values over other sports and even reference values [22,23,24,25,26], these results can be supported by previous research in endurance sports [2,9,19,20,21]. The absence of differences may not indicate a lack of specific adaptations to swim training; for example, all the runners in this sample had prior swim training, and some continued to use swim training as cross-training. After analyzing the results simply by practiced sport, the next step of this research was deepening the analysis of the results based on swimming years of experience and training onset.

### 4.2. Swim Training Onset Does Not Appear to Influence Lung Function

Years of swim training experience and lung function showed a moderate correlation with FVC (r = 0.337, *p* = 0.023 with z-score and r = 0.341, *p* = 0.022 with %RV) and FEV_1_ (r = 0.340, *p* = 0.022 with z-score and r = 0.347, *p* = 0.020 with %RV), but only when measured in the immersed seated condition. This may explain why other studies conducted in the on-land seated condition found no correlations between years of swimming experience and lung function [1,18] or heterogeneous results [4,11,19,32,37,38,39]. The adaptations to swim training might be more easily observed in measurements performed in the water, as the muscles and lungs are closer to the exposed pressure, which could explain the induced chronic training adaptations. Based on these results, the immersed seated condition in the aquatic environment appears to effectively reflect the adaptations to swim training, as it is more closely related to the training environment [40].

Interestingly, when the sample was divided by the timing of swim training onset, the ES and NES groups showed differences in FVC_PI_ using both z-score (*p* = 0.035) and %RV (*p* = 0.032), as well as in FVC_SI_, but only with %RV (*p* = 0.045). However, as expected, they also differed in ST_years_ (*p* = 0.011). After analyzing the differences with ST_years_ as a covariate, the differences in lung function between groups were no longer observable, which was also found in other studies for the on-land seated condition [1,4,18]. This suggests that accumulated years of swimming experience may lead to increased measurements of FVC and FEV_1_ in the immersed seated condition, and that these adaptations may not be exclusive to, nor more pronounced in, athletes who began swimming earlier in life. Longitudinal studies should be performed to analyze this possibility.

The results of this study do not support the notion that there is a critical window of opportunity during childhood for enhanced lung function adaptations through swimming training [1,4,13,18]. Nonetheless, this was only possible when analyzing group differences with ST_years_ as a covariate, as early starters always had more years of accumulated experience than the other athletes. Including this covariate made it possible to eliminate the confounding variable and isolate the differences based solely on swim training onset.

### 4.3. Effects of Swim Training in Allergic and Asthmatic Athletes

Looking at the sample division by RA and RAs, no differences between negative and positive reported allergy and reported asthma groups could be found in lung function or the accumulated years of swimming training. However, when correlations were performed, the NRA and NRAs groups showed a correlation between FVC_SI_ and accumulated years of swimming training (r = 0.469, *p* = 0.010 by the z-score and r = 0.468, *p* = 0.010 by %RV for the NRA group and r = 0.380, *p* = 0.022 by the z-score and r = 0.382, *p* = 0.022 by %RV for the NRAs group) (Table 4). In healthy athletes, the greater the years of swimming training, the greater the FVC_SI_. However, that does not seem to happen in male athletes with asthma due to the already increased FVC promoted by the disease [3,41,42,43,44,45], at least in the immersed seated condition.

The most interesting results were that, in the positive groups, moderate and strong correlations with the accumulated years of swimming training were observable with the FEV_1_/FVC (r = 0.606, *p* = 0.013 by the z-score and r = 0.589, *p* = 0.016 by %RV for the PRA group in the SL condition; r = 0.570, *p* = 0.021 by %RV for the PRA group in the PL condition; r = 0.850, *p* = 0.004 by the z-score and r = 0.833, *p* = 0.005 by %RV for the PRAs group in the SL condition; r = 0.767, *p* = 0.016 by the z-score and r = 0.710, *p* = 0.032 by %RV for the PRAs group in the PL condition; and r = 0.800, *p* = 0.010 by the z-score and r = 0.758, *p* = 0.018 by %RV for the PRAs group in the SI condition) and FEF_25–75%_ (r = 0.576, *p* = 0.020 by the z-score and r = 0.594, *p* = 0.015 by %RV for the PRA group in the SL condition; r = 0.583, *p* = 0.018 by the z-score and r = 0.624, *p* = 0.010 by %RV for the PRA group in the PL condition; r = 0.571, *p* = 0.021 by %RV for the PRA group in the SI condition; r = 0.555, *p* = 0.026 by %RV for the PRA group in the PI condition; r = 0.717, *p* = 0.030 by the z-score and r = 0.786, *p* = 0.012 by %RV for the PRAs group in the SL condition; and r = 0.673, *p* = 0.047 by %RV for the PRAs group in the PL condition) variables (Table 4). Allergic and asthmatic athletes could benefit from swim training, as it appears to be connected to enhanced lung function variables typically reduced in asthma: FEV_1_, FEV_1_/FVC, and FEF_25–75%_ [36,46,47]. Therefore, swim training could enhance different variables in healthy and allergic/asthmatic athletes, thereby mitigating the differences in lung function between groups and explaining the lack of differences between allergic/asthmatic athletes and their non-allergic counterparts observed in this study. Again, longitudinal studies should be performed to confirm this hypothesis.

## 5. Limitations

Due to sample limitations, this research only evaluated male athletes. Research in female athletes should be performed as well, as differences in both (1) the respiratory response to changes in environment and body position and (2) respiratory health can be significant. The software used for spirometric data collection did not provide a reference value for the PEF variable, so it was not included when calculating the z-score and the percentage of predicted reference values. Groups created for analyses were based on self-reported data from the questionnaires, and the lack of on-site medical evaluations could imply that some athletes might have some kind of undiagnosed allergy and/or asthma; however, this type of extensive evaluation was not possible under the circumstances of this research.

## 6. Conclusions

The conclusions of this study are threefold: First, the immersed seated condition appears to more accurately measure differences in lung function related to swim training experience. Second, the results do not support the notion of a critical window during childhood for enhanced lung function adaptations due to swim training. However, the accumulated years of swim training explain the increased FVC and FEV_1_ measurements in male athletes, particularly in the immersed seated condition. Lastly, swim training correlates with different lung function variables in healthy versus allergic/asthmatic male athletes when measured in the immersed seated condition. Specifically, healthy male athletes showed a positive trend in FVC, and allergic and asthmatic male athletes showed a positive trend in FEV_1_/FVC and FEF_25–75%_. More research appears to be needed in this area.

## Figures and Tables

**Table 1 jfmk-10-00231-t001:** Years of swimming experience and training session frequency weekly averages across sports.

	Swimmers	Triathletes	Runners	Test Value	Significance
ST_years_	22.93 ± 8.29 **^b^	13.07 ± 8.40	5.73 ± 8.29	H = 20.654	*p* < 0.001 **
TSFreq_Total_	4.60 ± 2.26	6.67 ± 2.87	5.27 ± 1.94	F = 2.922	*p* = 0.065
TSFreq_swim_	2.40 ± 0.91 **^a^	2.40 ± 0.83 **^c^	0.20 + 0.56	H = 27.995	*p* < 0.001 **

** *p* < 0.01; ^a^ difference between swimmers and triathletes; ^b^ difference between swimmers and runners; ^c^ difference between triathletes and runners; ST_years_—years of cumulative experience in formal swimming training; TSFreq_Total_—weekly total training session frequency; TSFreq_swim_—weekly swim training session frequency; H—Mann–Whitney test value; and F—ANOVA test value.

**Table 2 jfmk-10-00231-t002:** Lung function z-score and percentage of predicted reference value data across sports (based on GLI 2012-predicted reference values).

		Swimmers	Triathletes	Runners	Test Value	Significance
FVC_SL_	Z-score	0.91 ± 1.16	0.51 ± 0.78	0.62 ± 0.48	F = 0.872	*p* = 0.426
	%RV	111.53 ± 14.85	106.40 ± 10.16	108.00 ± 6.00	F = 0.863	*p* = 0.429
FVC_PL_	Z-score	0.21 ± 1.13	−0.13 ± 0.74	−0.05 ± 0.44	F = 0.714	*p* = 0.496
	%RV	102.67 ± 14.34	98.33 ± 9.66	99.40 ± 5.53	F = 0.697	*p* = 0.504
FVC_SI_	Z-score	−0.27 ± 1.17	−0.73 ± 0.66	−0.68 ± 0.64	F = 0.943	*p* = 0.288
	%RV	96.73 ± 14.78	90.60 ± 8.42	91.53 ± 7.68	F = 1.411	*p* = 0.255
FVC_PI_	Z-score	0.08 ± 1.34	−0.43 ± 0.66	−0.39 ± 0.60	F = 1.223	*p* = 0.254
	%RV	101.07 ± 17.02	94.40 ± 8.49	95.13 ± 7.49	F = 1.439	*p* = 0.249
FEV_1SL_	Z-score	0.83 ± 1.12	0.55 ± 0.73	0.30 ± 0.87	F = 1.226	*p* = 0.304
	%RV	110.13 ± 13.86	106.80 ± 9.07	103.53 ± 10.73	F = 1.258	*p* = 0.295
FEV_1PL_	Z-score	0.14 ± 1.10	−0.06 ± 0.65	−0.41 ± 0.74	F = 1.789	*p* = 0.180
	%RV	101.87 ± 14.02	99.13 ± 8.18	94.93 ± 7.60	F = 1.709	*p* = 0.193
FEV_1SI_	Z-score	−0.40 ± 1.29	−0.71 ± 0.53	−1.00 ± 0.74	F = 1.594	*p* = 0.215
	%RV	95.07 ± 16.24	90.93 ± 6.60	87.40 ± 9.45	F = 1.671	*p* = 0.200
FEV_1PI_	Z-score	0.04 ± 1.22	−0.43 ± 0.66	−0.58 ± 0.78	F = 1.838	*p* = 0.172
	%RV	100.60 ± 15.18	94.47 ± 8.35	92.67 ± 9.78	F = 1.967	*p* = 0.153
FEV_1_/FVC_SL_	Z-score	−0.10 ± 1.07	0.05 ± 0.80	−0.55 ± 0.94	F = 1.611	*p* = 0.212
	%RV	98.87 ± 8.04	100.27 ± 5.97	95.60 ± 7.48	F = 1.651	*p* = 0.204
FEV_1_/FVC_PL_	Z-score	−0.04 ± 1.26	0.15 ± 1.03	−0.61 ± 0.83	F = 2.085	*p* = 0.137
	%RV	99.27 ± 9.46	100.87 ± 7.58	95.07 ± 6.32	F = 2.161	*p* = 0.128
FEV_1_/FVC_SI_	Z-score	−0.22 ± 1.13	0.07 ± 0.88	−0.59 ± 0.89	H = 4.008	*p* = 0.135
	%RV	97.93 ± 8.71	100.27 ± 6.29	95.13 ± 6.77	H = 4.511	*p* = 0.105
FEV_1_/FVC_PI_	Z-score	0.02 ± 1.27	0.04 ± 1.02	−0.36 ± 0.85	F = 0.667	*p* = 0.519
	%RV	99.53 ± 9.70	99.93 ± 7.55	96.93 ± 6.55	F = 0.616	*p* = 0.545
FEF_25–75%SL_	Z-score	0.27 ± 0.97	0.20 ± 0.66	−0.23 ± 0.90	F = 1.483	*p* = 0.239
	%RV	111.00 ± 32.59	107.67 ± 20.46	94.53 ± 26.57	F = 1.560	*p* = 0.222
FEF_25–75%PL_	Z-score	0.05 ± 1.19	0.04 ± 0.81	−0.57 ± 0.75	F = 2.144	*p* = 0.130
	%RV	106.33 ± 40.33	103.20 ± 24.62	85.20 ± 21.63	F = 2.168	*p* = 0.127
FEF_25–75%SI_	Z-score	−0.29 ± 1.32	−0.24 ± 0.76	−0.85 ± 0.82	H = 5.257	*p* = 0.072
	%RV	97.60 ± 43.44	95.07 ± 21.35	78.27 ± 21.99	H = 5.848	*p* = 0.054
FEF_25–75%PI_	Z-score	0.02 ± 1.18	−0.16 ± 0.87	−0.51 ± 0.86	F = 1.130	*p* = 0.333
	%RV	105.53 ± 38.81	97.67 ± 25.39	87.47 ± 23.87	F = 1.357	*p* = 0.268

FVC—Forced Vital Capacity; FEV_1_—Forced Expiratory Volume in One Second; FEF—Forced Expiratory Flow; %RV—Percentage of Predicted Reference Values; SL—On-Land Seated Condition; PL—On-Land Prone Condition; SI—Immersed Seated Condition; PI—Immersed Prone Condition; H—Mann–Whitney Test Value; and F—ANOVA Test Value.

**Table 3 jfmk-10-00231-t003:** Correlations between years of swimming experience (ST_years_) and lung function variables.

Correlations		ST_years_
FVC_SL_	yz-score	r = 0.169; *p* = 0.266
	%RV	r = 0.164; *p* = 0.281
FVC_PL_	z-score	r = 0.213; *p* = 0.160
	%RV	r = 0.210; *p* = 0.166
FVC_SI_	z-score	r = 0.337; *p* = 0.023 *
	%RV	r = 0.341; *p* = 0.022 *
FVC_PI_	z-score	r = 0.212; *p* = 0.163
	%RV	r = 0.218; *p* = 0.149
FEV_1SL_	z-score	r = 0.278; *p* = 0.064
	%RV	r = 0.277; *p* = 0.066
FEV_1PL_	z-score	r = 0.277; *p* = 0.066
	%RV	r = 0.262; *p* = 0.082
FEV_1SI_	z-score	r = 0.340; *p* = 0.022 *
	%RV	r = 0.347; *p* = 0.020 *
FEV_1PI_	z-score	r = 0.239; *p* = 0.114
	%RV	r = 0.250; *p* = 0.097
FEV_1_/FVC_SL_	z-score	r = 0.083; *p* = 0.589
	%RV	r = 0.084; *p* = 0.583
FEV_1_/FVC_PL_	z-score	r = 0.035; *p* = 0.817
	%RV	r = 0.050; *p* = 0.745
FEV_1_/FVC_SI_	z-score	r = 0.004; *p* = 0.980
	%RV	r = 0.003; *p* = 0.987
FEV_1_/FVC_PI_	z-score	r = −0.019; *p* = 0.904
	%RV	r = −0.009; *p* = 0.954
FEF_25–75%SL_	z-score	r = 0.172; *p* = 0.259
	%RV	r = 0.179; *p* = 0.239
FEF_25–75%PL_	z-score	r = 0.117; *p* = 0.443
	%RV	r = 0.103; *p* = 0.501
FEF_25–75%SI_	z-score	r = 0.149; *p* = 0.329
	%RV	r = 0.153; *p* = 0.315
FEF_25–75%PI_	z-score	r = 0.095; *p* = 0.536
	%RV	r = 0.087; *p* = 0.572

* *p* < 0.05; FVC—Forced Vital Capacity; FEV_1_—Forced Expiratory Volume in One Second; FEF—Forced Expiratory Flow; %RV—Percentage of Predicted Reference Values; SL—On-Land Seated Condition; PL—On-Land Prone Condition; SI—Immersed Seated Condition; and PI—Immersed Prone Condition.

**Table 4 jfmk-10-00231-t004:** Correlations between years of swimming experience and allergy diagnosis and asthma diagnosis groups.

Correlations		ST_years_			
		NRA	NRAs	PRA	PRAs
FVC_SL_	z-score	r = 0.348; *p* = 0.065	r = 0.241; *p* = 0.157	r = −0.057; *p* = 0.833	r = −0.243; *p* = 0.529
	%RV	r = 0.349; *p* = 0.064	r = 0.238; *p* = 0.163	r = −0.081; *p* = 0.766	r = 0.070; *p* = 0.859
FVC_PL_	z-score	r = 0.296; *p* = 0.119	r = 0.258; *p* = 0.128	r = 0.050; *p* = 0.854	r = −0.350; *p* = 0.356
	%RV	r = 0.295; *p* = 0.120	r = 0.254; *p* = 0.135	r = 0.048; *p* = 0.859	r = −0.063; *p* = 0.873
FVC_SI_	z-score	r = 0.469; *p* = 0.010 *	r = 0.380; *p* = 0.022 *	r = 0.062; *p* = 0.820	r = 0.133; *p* = 0.732
	%RV	r = 0.468; *p* = 0.010 *	r = 0.382; *p* = 0.022 *	r = 0.069; *p* = 0.800	r = 0.163; *p* = 0.676
FVC_PI_	z-score	r = 0.320; *p* = 0.091	r = 0.314; *p* = 0.062	r = 0.063; *p* = 0.818	r = −0.367; *p* = 0.322
	%RV	r = 0.333; *p* = 0.078	r = 0.319; *p* = 0.058	r = 0.058; *p* = 0.832	r = −0.082; *p* = 0.833
FEV_1SL_	z-score	r = 0.224; *p* = 0.244	r = 0.226; *p* = 0.185	r = 0.406; *p* = 0.119	r = 0.650; *p* = 0.058
	%RV	r = 0.242; *p* = 0.206	r = 0.222; *p* = 0.193	r = 0.412; *p* = 0.13	r = 0.590; *p* = 0.094
FEV_1PL_	z-score	r = 0.185; *p* = 0.337	r = 0.233; *p* = 0.171	r = 0.467; *p* = 0.068	r = 0.400; *p* = 0.286
	%RV	r = 0.144; *p* = 0.456	r = 0.203; *p* = 0.235	r = 0.485; *p* = 0.057	r = 0.449; *p* = 0.225
FEV_1SI_	z-score	r = 0.297; *p* = 0.117	r = 0.274; *p* = 0.106	r = 0.301; *p* = 0.257	r = 0.350; *p* = 0.356
	%RV	r = 0.298; *p* = 0.117	r = 0.280; *p* = 0.098	r = 0.327; *p* = 0.216	r = 0.450; *p* = 0.225
FEV_1PI_	z-score	r = 0.147; *p* = 0.446	r = 0.203; *p* = 0.235	r = 0.350; *p* = 0.184	r = 0.217; *p* = 0.576
	%RV	r = 0.167; *p* = 0.388	r = 0.220; *p* = 0.197	r = 0.372; *p* = 0.156	r = 0.209; *p* = 0.589
FEV_1_/FVC_SL_	z-score	r = −0.164; *p* = 0.395	r = −0.058; *p* = 0.736	r = 0.606; *p* = 0.013 *	r = 0.850; *p* = 0.004 **
	%RV	r = −0.157; *p* = 0.415	r = −0.053; *p* = 0.761	r = 0.589; *p* = 0.016 *	r = 0.833; *p* = 0.005 **
FEV_1_/FVC_PL_	z-score	r = −0.225; *p* = 0.241	r = −0.104; *p* = 0.545	r = 0.412; *p* = 0.112	r = 0.767; *p* = 0.016 *
	%RV	r = −0.206; *p* = 0.284	r = −0.083; *p* = 0.628	r = 0.570; *p* = 0.021 *	r = 0.710; *p* = 0.032 *
FEV_1_/FVC_SI_	z-score	r = −0.198; *p* = 0.304	r = −0.104; *p* = 0.545	r = 0.481; *p* = 0.059	r = 0.800; *p* = 0.010 *
	%RV	r = −0.181; *p* = 0.348	r = −0.099; *p* = 0.564	r = 0.409; *p* = 0.116	r = 0.758; *p* = 0.018 *
FEV_1_/FVC_PI_	z-score	r = −0.218; *p* = 0.257	r = −0.117; *p* = 0.496	r = 0.480; *p* = 0.060	r = 0.417; *p* = 0.265
	%RV	r = −0.194; *p* = 0.314	r = −0.106; *p* = 0.538	r = 0.479; *p* = 0.061	r = 0.516; *p* = 0.155
FEF_25–75%SL_	z-score	r = −0.046; *p* = 0.814	r = 0.027; *p* = 0.875	r = 0.576; *p* = 0.020 *	r = 0.717; *p* = 0.030 *
	%RV	r = −0.033; *p* = 0.866	r = 0.031; *p* = 0.857	r = 0.594; *p* = 0.015 *	r = 0.786; *p* = 0.012 *
FEF_25–75%PL_	z-score	r = −0.130; *p* = 0.502	r = −0.006; *p* = 0.972	r = 0.583; *p* = 0.018 *	r = 0.533; *p* = 0.139
	%RV	r = −0.147; *p* = 0.445	r = −0.023; *p* = 0.893	r = 0.624; *p* = 0.010 *	r = 0.673; *p* = 0.047 *
FEF_25–75%SI_	z-score	r = −0.084; *p* = 0.666	r = 0.018; *p* = 0.915	r = 0.407; *p* = 0.118	r = 0.427; *p* = 0.252
	%RV	r = −0.085; *p* = 0.661	r = 0.023; *p* = 0.896	r = 0.571; *p* = 0.021 *	r = 0.577; *p* = 0.104
FEF_25–75%PI_	z-score	r = −0.111; *p* = 0.566	r = 0.008; *p* = 0.963	r = 0.497; *p* = 0.050	r = 0.317; *p* = 0.406
	%RV	r = −0.139; *p* = 0.471	r = −0.004; *p* = 0.982	r = 0.555; *p* = 0.026 *	r = 0.472; *p* = 0.199

* *p* < 0.05; ** *p* < 0.01; FVC—forced vital capacity; FEV_1_—forced expiratory volume in one second; FEF—forced expiratory flow; %RV—percentage of predicted reference values; SL—on-land seated condition; PL—on-land prone condition; SI—immersed seated condition; PI—immersed prone condition; NRA—negative reported physician-diagnosed allergy or asthma; NRAs—negative reported physician-diagnosed asthma; PRA—positive reported physician-diagnosed allergy or asthma; and PRAs—positive reported physician-diagnosed asthma.

**Table 5 jfmk-10-00231-t005:** Lung function z-score and percentage of predicted reference value data in groups with different swim training start onset times (based on GLI 2012-predicted reference values). Analysis of co-variance performed with years of swim training experience (ST_years_) as the co-variable.

		ES	NES	Test Value	Co-variance Test
ST_years_		17.30 ± 10.46	8.83 ± 9.48	H = −2.540, *p* = 0.011 *	N.A.
FVC_SL_	Z-score	0.87 ± 0.93	0.39 ± 0.64	F = 1.931, *p* = 0.060	F = 1.704, *p* = 0.199
	%RV	111.22 ± 11.83	104.78 ± 8.24	F = 2.006, *p* = 0.051	F = 1.892, *p* = 0.176
FVC_PL_	Z-score	0.15 ± 0.89	−0.19 ± 0.66	F = 1.415, *p* = 0.164	F = 0.531, *p* = 0.470
	%RV	101.96 ± 11.32	97.39 ± 8.43	F = 1.463, *p* = 0.151	F = 0.612, *p* = 0.438
FVC_SI_	Z-score	−0.36 ± 0.97	−0.86 ± 0.58	F = 1.963, *p* = 0.056	F = 1.122, *p* = 0.295
	%RV	95.59 ± 12.13	89.00 ± 7.32	F = 2.065, *p* = 0.045 *	F = 1.330, *p* = 0.255
FVC_PI_	Z-score	−0.03 ± 1.07	−0.57 ± 0.58	F = 2.179, *p* = 0.035 *	F = 1.517, *p* = 0.225
	%RV	99.67 ± 13.54	92.67 ± 7.49	F = 2.224, *p* = 0.032 *	F = 1.598, *p* = 0.213
FEV_1SL_	Z-score	0.67 ± 1.01	0.39 ± 0.78	F = 1.008, *p* = 0.319	F = 0.101, *p* = 0.752
	%RV	108.19 ± 12.50	104.78 ± 9.68	F = 0.976, *p* = 0.334	F = 0.077, *p* = 0.783
FEV_1PL_	Z-score	−0.08 ± 0.94	−0.16 ± 0.66	F = 0.298, *p* = 0.767	F = 0.223, *p* = 0.639
	%RV	99.04 ± 11.90	98.06 ± 8.30	F = 0.304, *p* = 0.763	F = 0.216, *p* = 0.644
FEV_1SI_	Z-score	−0.59 ± 1.09	−0.87 ± 0.60	F = 1.013, *p* = 0.317	F = 0.034, *p* = 0.854
	%RV	92.67 ± 13.58	88.83 ± 7.81	F = 1.081, *p* = 0.286	F = 0.053, *p* = 0.818
FEV_1PI_	Z-score	−0.19 ± 1.05	−0.51 ± 0.72	F = 1.129, *p* = 0.265	F = 0.272, *p* = 0.605
	%RV	97.59 ± 13.05	93.39 ± 9.21	F = 1.008, *p* = 0.319	F = 0.302, *p* = 0.585
FEV_1_/FVC_SL_	Z-score	−0.32 ± 0.97	−0.01 ± 0.92	F = −1.183, *p* = 0.244	F = 1.787, *p* = 0.188
	%RV	97.26 ± 7.49	99.72 ± 7.01	F = −1.108, *p* = 0.274	F = 1.922, *p* = 0.173
FEV_1_/FVC_PL_	Z-score	−0.35 ± 1.09	0.11 ± 1.04	F = −1.394, *p* = 0.171	F = 2.503, *p* = 0.121
	%RV	97.04 ± 8.25	100.44 ± 7.63	F = −1.398, *p* = 0.169	F = 2.542, *p* = 0.118
FEV_1_/FVC_SI_	Z-score	−0.41 ± 1.01	−0.01 ± 0.94	H = 1.448, *p* = 0.148	H = −1.478, *p* = 0.147
	%RV	96.52 ± 7.80	99.67 ± 6.71	H = 1.590, *p* = 0.112	H = −1.626, *p* = 0.111
FEV_1_/FVC_PI_	Z-score	−0.25 ± 1.07	0.12 ± 1.02	F = −1.136, *p* = 0.262	F = 1.205, *p* = 0.278
	%RV	97.67 ± 8.21	100.50 ± 7.49	F = −1.173, *p* = 0.247	F = 1.292, *p* = 0.262
FEF_25–75%SL_	Z-score	0.06 ± 0.91	0.11 ± 0.80	F = −0.180, *p* = 0.858	F = 0.407, *p* = 0.527
	%RV	103.81 ± 29.63	105.28 ± 24.30	H = 0.487, *p* = 0.627	H = −0.956, *p* = 0.344
FEF_25–75%PL_	Z-score	−0.26 ± 1.02	0.00 ± 0.87	H = 1.413, *p* = 0.158	H = −1.764, *p* = 0.085
	%RV	95.63 ± 33.72	102.17 ± 26.23	H = 1.367, *p* = 0.171	H = −1.671, *p* = 0.102
FEF_25–75%SI_	Z-score	−0.53 ± 1.12	−0.36 ± 0.85	H = 1.205, *p* = 0.228	H = −1.631, *p* = 0.110
	%RV	89.26 ± 35.94	91.89 ± 23.28	H = 1.345, *p* = 0.179	H = −1.798, *p* = 0.079
FEF_25–75%PI_	Z-score	−0.26 ± 1.02	−0.14 ± 0.94	F = −0.412, *p* = 0.683	F = 0.452, *p* = 0.505
	%RV	95.78 ± 32.58	98.56 ± 27.51	H = 0.707, *p* = 0.480	H = −0.929, *p* = 0.358

* *p* < 0.05; FVC—forced vital capacity; FEV1—forced expiratory volume in one second; FEF—forced expiratory flow; %RV—percentage of predicted reference values; SL—on-land seated condition; PL—on-land prone condition; SI—immersed seated condition; PI—immersed prone condition; ES—early swim training onset; NES—non-early swim training onset; and N.A.—not applicable.

## Data Availability

Data used to produce the results of this article has been made accessible in a Mendeley Data database: DOI: 10.17632/rn2h8y8354.1.

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
