# Peer review of "Lung Function Changes with Swim Training in Healthy and Allergic Endurance Athletes"

_jfmk, 2025, doi:10.3390/jfmk10020231_

Round 1
Reviewer 1 Report
Comments and Suggestions for Authors
It is well established that lung capacity is of high importance in sports. From this point of view, knowing how different types of exercises influence and are influenced by lung function is interesting and vital. Swimming is believed to be the sport discipline influencing the lung functions the most. The Authors characterized athletes' lung function according to their swimming experience. The study is well designed and performed. However, I have some minor comments I’d like to express.
- One of the steps of data analysis was the subdivision according to the asthma. There is no information if asthmatic participants were taking any medications known to affect lung parameters and thus, the results of the study.
- Similarly, did the allergic athletes take any medications that possibly may influence the test results?
- The accuracy of some data, namely the participants age and height seem to be a little confusing. I understand that the data are mean ± SD. However, taking into account that 1/100 of the year equals about 3 days, it is hard to believe that the age was given in the accuracy of days.
- Giving p = 0.0000 in my opinion should be replaced by p < 0.0001.
The text of the manuscript should be carefully proof-read for typographic mistakes (e.g. unnecessary full stops).
Author Response
Comment 1: " There is no information if asthmatic participants were taking any medications known to affect lung parameters and thus, the results of the study. Similarly, did the allergic athletes take any medications that possibly may influence the test results?"
Reply 1: Thank you for pointing this out, as this should have been included in the procedures section. The medication control was instructed to the participants, however, its writing was missing in the paper. The following sentence was added to the Spirometry procedures section: "Athletes were instructed not to take any asthma/allergy medication 12 hours before data collection to prevent their effects from interfering with lung function variables."
Comment 2: " The accuracy of some data, namely the participants age and height seem to be a little confusing. I understand that the data are mean ± SD. However, taking into account that 1/100 of the year equals about 3 days, it is hard to believe that the age was given in the accuracy of days."
Reply 2: We gladly explain how we were able to get decimal data for participants' age: We used COSMED Omnia software, which requires the patient's birthday. From that is possible to obtain decimal data for age, which we report in the manuscript.
Comment 3: "Giving p = 0.0000 in my opinion should be replaced by p < 0.0001."
Reply 3: Thank you again for pointing this out, as it is a typo. The data was taken directly from SPSS and should have been rewritten. All those typos were replaced.
Comment 4: "The text of the manuscript should be carefully proof-read for typographic mistakes (e.g. unnecessary full stops)."
Reply 4: This is a valuable remark. The text has been carefully revised for typographic errors.
Reviewer 2 Report
Comments and Suggestions for Authors
Thank you for the opportunity to evaluate this scientific article. Specifically, my suggestions to the authors are the following:
1. Introduction
The lack of a clear operationalized hypothesis is noticeable (Lines:71–75)
It can be noted that the objectives are presented as general questions, without the explicit formulation of testable hypotheses.
The authors should formulate clear, specific hypotheses related to each objective (e.g., “We expect FVC to be significantly higher in swimmers than in runners under immersion conditions”).
2. Materials and Methods
The exclusive selection of male participants without control for bias (Lines:106–109) highlights the exclusion of women as being justified only by “differences in allergic response” and small numbers – but without analyzing the impact of this decision on external validity. I believe that the authors should include a discussion of the possible limitations of the generalization of the results.
3. Results
There is no significant difference between sports – poor interpretation - Lines: 226–229 and 244–249. The paragraphs state the lack of differences between sports without further analysis for possible confounders (e.g. age, allergy level, swimming history among runners). I believe that additional ANCOVA analysis or segmentation by previous exposure to swimming would be useful.
4. Discussion
There is a confusion between correlation and causation (lines: 319–321, 351–354). Clarification/rewording is needed to correctly reflect the correlative nature of the data as causal conclusions are drawn (e.g. “swimming training increases FVC”) without specifying that this is a cross-sectional, not longitudinal study.
5. Conclusions
Generalizations are found beyond the limits of the sample
(lines: 362–371), stating clear differences between healthy and allergic/asthmatic athletes without adequate statistical power for such small subgroups. It would be more correct to mention these as "observed trends" and not firm generalizations.
Author Response
Comment 1: "1. Introduction. The lack of a clear operationalized hypothesis is noticeable (Lines:71–75)
It can be noted that the objectives are presented as general questions, without the explicit formulation of testable hypotheses. The authors should formulate clear, specific hypotheses related to each objective (e.g., “We expect FVC to be significantly higher in swimmers than in runners under immersion conditions”)."
Reply 1: Thank you for this remark. The Introduction section was updated with the following sentences: "It was hypothesized that the years of swimming experience could be related to increases in FVC and FEV1, especially in the prone immersed condition, and that athletes who started swimming during childhood would present greater FVC and FEV1 than athletes who started swimming as adults. We also hypothesized that asthmatics, but not allergic or healthy athletes, would present lower values of FEV1 and FEF25-75% out of the water, however, with the help of hydrostatic pressure, these values would be soothed."
Comment 2: "2. Materials and Methods The exclusive selection of male participants without control for bias (Lines:106–109) highlights the exclusion of women as being justified only by “differences in allergic response” and small numbers – but without analyzing the impact of this decision on external validity. I believe that the authors should include a discussion of the possible limitations of the generalization of the results."
Reply 2: Thank you once again for your very pertinent comment. You are correct that the exclusivity of the results to male athletes was not sufficiently emphasized in the original manuscript. To address this, we revised the Discussion and Conclusion sections to replace general references to 'athletes' with 'male athletes' where appropriate. Additionally, the sentence in the Limitations section has been updated to more clearly emphasize that the findings apply exclusively to male athletes
Comment 3: "There is no significant difference between sports – poor interpretation - Lines: 226–229 and 244–249. The paragraphs state the lack of differences between sports without further analysis for possible confounders (e.g. age, allergy level, swimming history among runners). I believe that additional ANCOVA analysis or segmentation by previous exposure to swimming would be useful."
Reply 3: We appreciate the concern of the reviewer and understand the comment about the poor interpretation of the results. We did it so we could compare it with previous results. As recommended in the comment, segmentation by previous exposure to swim training was performed (as it had been in past literature), and an ANCOVA analysis was conducted, which was one of the novel aspects of this research.
Due to the extensive data, we rearranged the Discussion section to make it easier for readers to understand the steps of the research. The section "Lung Function differences between swim training onset" was moved to section 4.2. and "Effects of Asthma and Allergy" was switched to section 4.3.
Comment 4: "There is a confusion between correlation and causation (lines: 319–321, 351–354). Clarification/rewording is needed to correctly reflect the correlative nature of the data as causal conclusions are drawn (e.g. “swimming training increases FVC”) without specifying that this is a cross-sectional, not longitudinal study."
Reply 4: We appreciate your very pertinent comment once again. The discussion and conclusion were thoroughly revised to reflect correlations rather than causation.
Comment 5: "Generalizations are found beyond the limits of the sample
(lines: 362–371), stating clear differences between healthy and allergic/asthmatic athletes without adequate statistical power for such small subgroups. It would be more correct to mention these as "observed trends" and not firm generalizations."
Reply 5: Following your pertinent recommendations, the Conclusion section was rewritten to eliminate generalizations, such as not referencing male athletes and stating apparent differences solely based on the results of this paper.
Reviewer 3 Report
Comments and Suggestions for Authors
There appears to be a need for this research. The introduction concluded with a clear purpose statement. The methods section is well organized but needs additional information. The results section needs to be clarified in some sections. The results, discussion, and conclusion relate to the primary purpose. More specific information is needed in the discussion related to the strength of correlations. There are several word tense, word choice, and grammatical errors that need to be corrected. See specific comments in the pdf.

Author Response
We would like to express our gratitude for all the comments of the reviewer, which were very helpful in improving the quality of the manuscript. We send back the PDF document with the replies to the comments (Please see the attachment), and have uploaded the Word manuscript with all the changes performed.

Round 2
Reviewer 1 Report
Comments and Suggestions for Authors
The authors have revised the manuscript and it may be accepted for publication in its current form.
Author Response
We thank you once again for your important input.
Reviewer 2 Report
Comments and Suggestions for Authors
I consider that, in the version provided by the authors, the paper is appropriate and can be accepted for publication.
Author Response

(The authors gave the same response as above.)

Reviewer 3 Report
Comments and Suggestions for Authors
The authors adequately addressed previous issues. Some remaining comments/questions need attention. See specific comments in the revised pdf.

Author Response
We thank you once again for your valuable comments. As before, the manuscript was uploaded with the changes highlighted. Please see the pdf attachment, which includes the replies to the comments.
